



# HydroGFD3.0: a 25 km global near real-time updated precipitation and temperature data set

Peter Berg[1], Fredrik Almén[1], and Denica Bozhinova[1]

[1]Swedish Meteorological and Hydrological Institute, Folkborgsvägen 17, 601 76 Norrköping, Sweden

**Correspondence:** Peter Berg (peter.berg@smhi.se)

**Abstract.** HydroGFD (Hydrological Global Forcing Data) is a data set of bias adjusted reanalysis data for daily precipitation, and minimum, mean, and maximum temperature. It is mainly intended for large scale hydrological modeling, but is also suitable for other impact modeling. The data set has an almost global land area coverage, excluding the Antarctic continent, at a horizontal resolution of $0.25°$, i.e. about 25 km. It is available for the complete ERA5 reanalysis time period; currently 1979 until five days ago. This period will be extended back to 1950 once the back catalogue of ERA5 is available. The historical period is adjusted using global gridded observational data sets, and to acquire real-time data, a collection of several reference data sets is used. Consistency in time is attempted by relying on a background climatology, and only making use of anomalies from the different data sets. Precipitation is adjusted for mean bias as well as the number or wet days in a month. The latter is relying on a calibrated statistical method with input only of the monthly precipitation anomaly, such that no additional input data about the number of wet days is necessary. The daily mean temperature is adjusted toward the monthly mean of the observations, and applied to 1 h timesteps of the ERA5 reanalysis. Daily mean, minimum and maximum temperature are then calculated. The performance of the HydroGFD3 data set is on par with other similar products, although there are significant differences in different parts of the globe, especially where observations are uncertain. Further, HydroGFD3 tends to have higher precipitation extremes, partly due to its higher spatial resolution. In this paper, we present the methodology, evaluation results, and how to access to the data set at doi:10.5281/zenodo.3871707.

## 1 Introduction

Precipitation ($P$) and temperature ($T$) are key driving parameters for many impact models, and there are now many observational data sets available. They differ regarding the spatio-temporal resolution, the historical coverage, and the data sources included in the product. However, when it comes to continuously updated near real-time data sets, there are very few available data sets. It is therefore challenging to find a product suitable for monitoring and initialization of forecasts for an impact model, i.e. a product that fulfills both a long historical period for calibration and validation, as well as real-time updates.

While most data sets now offer a rather long historical period, the real-time availability is a greater challenge. Merged satellite and gauge data sets such as CHIRPS (Funk et al., 2015a), CMORPH (Joyce et al., 2004), and PERSIANN-CDR (Ashouri et al., 2015) offer both high resolution and near-realtime components, but are limited to between the +/-50 or +/-60 degree latitude bands. Several data sets have made use of reanalysis data as a basis, adjusted using various gridded observational



data sets (Weedon et al., 2011, 2014; Beck et al., 2017; Berg et al., 2018). The advantage is that the reanalysis products are
readily available with a large range of variables and output frequencies. Still, the downside with reanalysis products is that
especially $P$ is a model product and thereby suffers from model bias. Since the bias can be substantial, several methods have
been developed to adjust reanalysis, using different methods and reference data sets.

30       A hydrological operational monitoring or forecast product has strong demands on availability and redundancy of the data
flows. The data set HydroGFD1 (Berg et al., 2018) was constructed and made operational for initializations of the hydrological
model HYPE (Lindström et al., 2010) for different set-ups across the globe. It offered near-realtime updating of daily $P$ and
daily $T$ (mean, minimum, and maximum), until the end of the last calendar month. The real-time components of HydroGFD1
were based on ERA-Interim reanalysis, extended by the ECMWF deterministic forecasts, adjusted using monthly mean $P$
from GPCC-Monitoring and GPCC-FirstGuess (Schneider et al., 2018b) products, and monthly mean $T$ from GHCN-CAMS
(Fan and Van den Dool, 2008). The follow-up data set HydroGFD2 offered some updates to the methodology, and shifted
to using primarily the CPC-Unified (Chen et al., 2008) and CPC-Temp (CPCtemp, 2017) products for $P$ and $T$ adjustments,
respectively. Both data sets have been operationally produced for a few years now, and we have identified some serious issues
regarding the availability of required data sets for successful updates. The largest operational intermission occurred during the
government lock-down in the US between the $22^{nd}$ of December 2018 and the $25^{th}$ of January 2019. Neither of the US data
sets included in the production were available, which hampered the production of the HydroGFD data sets, and subsequently
deteriorated the quality of some operational HYPE models. Both these HydroGFD versions have now become obsolete for
real-time production due to the discontinuation of the ERA-Interim production as of August 2019. Data sets using multiple
input data sources are less sensitive to such conditions, such as the MSWEP data set (Beck et al., 2017).

45       In this paper, the HydroGFD3.0 system is described, with its range of produced data sets for the period 1979 to near real-
time, at $0.25°$ resolution and global land coverage. We describe the methodology and the operational production, as well as an
evaluation of the climatological data set, with comparison to other similar data sources.

## 2   Data

Table 1 lists the data sources used in the production of the different *tiers* (see Methods section) of HydroGFD3. From now
on, we will use the shortened internal abbreviations listed under "Name" in Tab. 1 when we refer to the data of $P$ or $T$ from
each source. ERA5 is the latest global reanalysis product of the ECMWF (Hersbach et al., 2020) and forms the basis for
HydroGFD3. This reanalysis product is chosen because our operational forecasts at SMHI are based on the medium range
forecasts of ECMWF, with a similar model as that used for ERA5. Other reanalysis products would be possible, but are not
explored here. ERA5 is updated with a three months lag, but a new temporary product, ERA5T, is produced with a five days
lag.

56       The HydroGFD3 background climatology is based on cpct for $T$, and CHPclim (Funk et al., 2015b) and gpcch (Schneider
et al., 2018a) for $P$. For the historical period, HydroGFD3 utilizes the data set CRUts4.03 (Harris and Jones, 2019) for $T$ and
wet day frequency, and gpcch for $P$. Several tiers are performed for the near real-time updating using different input data sets,





**Table 1.** Table of model and data sources used in the production of HydroGFD3, as well as the WFDE5 data set used for comparison. Note the lower case abbreviations used in the main text and in figures which follow the internal notation used in the data set production.

| Data set | Name | Variables | Resolution | Period | Reference |
|---|---|---|---|---|---|
| ERA5 | e5 | $T, P$ | hourly; 0.33° | 1979–(t-3 months) | Hersbach et al. (2020) |
| ERA5T | e5t | $T, P$ | hourly; 0.33° | (t-3 months) – (t-5 days) | Hersbach et al. (2020) |
| CRUts4.03 | cru | $T, P, N_{wet}$ | monthly; 0.5° | 1901–(t-2 months) | Harris and Jones (2019) |
| GPCCv8 | gpcch | $P$ | monthly; 0.25° | 1891–2016 | Schneider et al. (2018a) |
| GPCC-monitoringv6 | gpccm | $P$ | monthly; 1.0° | 1982–(t-3 months) | Schneider et al. (2018b) |
| GPCC-First guess | gpccf | $P$ | monthly; 1.0° | 2004–(t-1 month) | Schneider et al. (2018b) |
| CPC-Unified | cpcp | $P$ | daily; 0.5° | 1979–(t-2 days) | Chen et al. (2008) |
| CPC-Temp | cpct | $T_{min}, T_{max}$ | daily; 0.5° | 1979–(t-2 days) | CPCtemp (2017) |
| CHPclimv1.0 | chpclim | $P$ | climatology; 0.05° | (1980–2009) | Funk et al. (2015b) |
| WFDE5-CRU | wfd-cru | $T, P$ | hourly; 0.5° | 1979-2018 | Cucchi et al. (2020) |
| WFDE5-GPCC | wfd-gpcc | $P$ | hourly; 0.5° | 1979-2016 | Cucchi et al. (2020) |

for redundancy. For $P$, there are products based on gpccm (Schneider et al., 2018b), gpccf (Schneider et al., 2018b), cpcp
(Chen et al., 2008), as well as a climatological adjustment. For $T$, there is only the cpct data set (CPCtemp, 2017), and the
climatological adjustment.
The analysis is comparing the different data sets included in the processing, and additionally makes use of the latest version
of the *WATCH forcing data* WFDE5 (Cucchi et al., 2020) as a state-of-the-art comparison.

## 3  Method

The main method that HydroGFD is building on consists of adding observational monthly anomalies to a background clima-
tology, then adjusting the reanalysis data to that absolute monthly mean. Further steps assure consistency between different
versions of the data set, e.g. regarding spatial coverage. The different steps in producing the HydroGFD3 data sets are presented
in detail in the following sections.

### 3.1  Climatology

The $P$ background climatology is based on the CHPclim high resolution climatology of satellite, gauge, and physiographic
indicators (Funk et al., 2015b). We retain the same climatological period throughout the HydroGFD3 data set. CHPclim comes
in two versions, one with full coverage for the 50°S–50°N latitude band, and one with global land coverage. We choose to
make the global coverage version the main choice, but add information from the tropical full coverage version to increase
coverage along coastlines and islands. The original 0.05° resolution is remapped conservatively to the 0.25° resolution of
the HydroGFD3 dataset. Some issues with the CHPclim data set were identified, with observational artifacts in mid-northern





Siberia, and underestimation in Scandinavia. Therefore, these two regions were replaced by GPCCv8 climatological data for
the 1980–2009 period (see supplementary material for details). To avoid introducing sharp boarders, a zone of five grid points
were used around each area as a linear transition from one data set to another. Since Greenland $P$ is poorly mapped by both
satellite and gauge data, we have chosen to let its climatology be defined by e5, rather than any of the data sets.
For $T$, we use the cpct climatology (1980–2009) with only a remapping to the 0.25 degree resolution, and in-filling of
missing data points using e5. The third climatology consists of the wet day frequency (1980–2009), which is taken from the
CRUts4.03 data set of gridded station observations of the number of wet days in a month.
In a final step, the three climatologies are harmonized by only retaining the grid points that are available consistently in all
data sets and all months. This leads also to the final land-mask of the HydroGFD3 data set.
The elevation is defined by the e5 surface geopotential divided by the gravity of Earth (9.80665 m/s$^2$).

## 3.2 Anomaly method

HydroGFD3 makes use of several different data sets, which need to be stitched together in different configurations depending
on the use. Without some kind of homogenization between the data sets, sharp changes in the data are unavoidable when
switching from one data set to another. The homogenization used here is performed by only making use of anomalies from the
different data sets.
In the earlier version HydroGFD1 (Berg et al., 2018), which is closely based on the WFD method (Weedon et al., 2011), each
month of the reanalysis data set is adjusted with the absolute monthly mean of the observational data set. This main principle is
retained, however, in a new homogenization step we create new absolute observations by first calculating the monthly anomaly
compared to the 1989–2009 climatological period calculated for each data set, then adding this anomaly to the HydroGFD3
climatology. Anomalies are additive for $T$
$$T_{anom}(year, month) = T(year, month) - T_{clim}(month) \qquad (1)$$
and multiplicative for $P$
$$P_{anom}(year, month) = P(year, month)/P_{clim}(month) \qquad (2)$$
The reverse operation is applied after replacing the climatology.

## 3.3 Wetday frequency

A common issue with coarse resolution models, such as e5, is a tendency to produce excessive drizzle that reduces the number
of dry days in a month. To alleviate potential excessive drizzle, the number of wet days are adjusted before correcting the $P$
amount. The wetday frequencies in a month are not well covered by observational monitoring records and the uncertainties are
large when available. We have chosen to estimate the number of wet days based on the method of Stillman and Zeng (2016).
The method essentially relates the number of wet days, $N_{wet}$, to the monthly $P$ anomaly, $P_{anom}$, using also the climatological



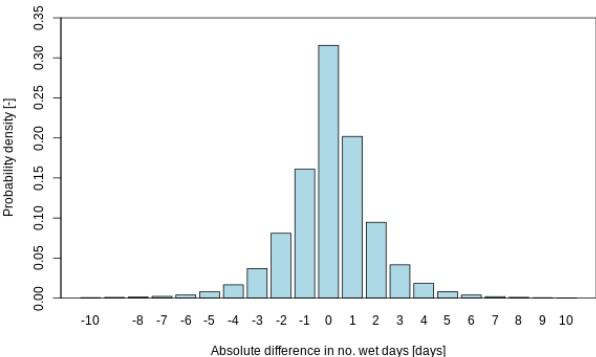

**Figure 1.** Distribution of the absolute difference in the number of wet days $N_{wet}$ estimated through the Stillman and Zeng (2016) method, and gpcch $P$. The probability density function ranges globally over all land grid points.

wet day frequency, $N_{wet}^{clim}$ as a predictor, and a tunable constant, $k$.
$N_{wet} = P_{anom}^{k} * N_{wet}^{clim}$                                                                     (3)
A value of $k = 0.28$ was derived for HydroGFD2.0 by calibration to the cru observations of the number of wet days in a
month, together with the cpcp $P$ observations. This value is almost half of that found by Stillman and Zeng (2016), which can
probably be related to the data sets used, but was found to be well applicable across the world. A verification of this constant
was performed with the cru wet days and the gpcch monthly $P$ anomalies, see Fig.1. This reveals an overall high accuracy of
the method, with deviations from observations of mostly only few days in a month, but can in rare cases be as much as ten
days. On average over the 1980–2009 period, and for each single grid point, the deviations are close to zero. Thus, the method
works well across all areas, and with sufficient precision for our purposes.
**3.4    Applied corrections**
The production of the corrected data consists of the following steps.
1. Calculate observed anomalies
2. Construct absolute reference data by adding the anomalies to the HydroGFD3 climatology
3. ($P$ only) Calculate the number of wet days
4. ($P$ only) Remove the weakest excessive wet days in e5
5. Calculate the ratio between the monthly means of the reference and e5
6. Apply the ratio to all time steps of e5





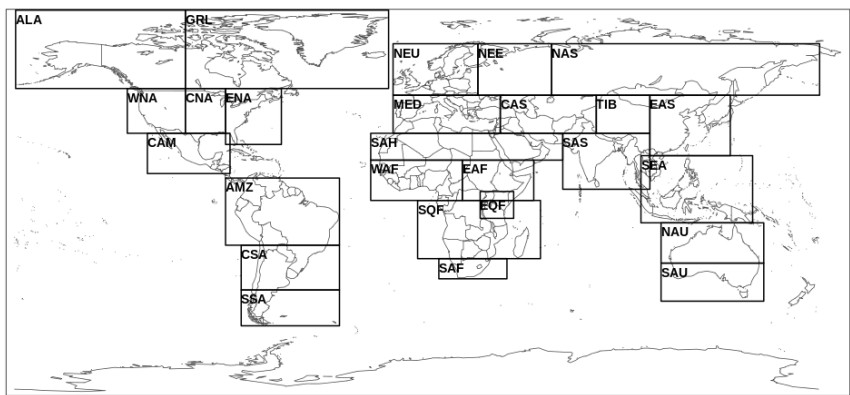

**Figure 2.** Evaluation regions as defined by Giorgi and Bi (2005), and employed in the PDF and time series analysis.

7. ($T$ only) Calculate mean, minimum, and maximum $T$ from the hourly time steps
For $P$, the scaling can cause very large values in some cases, e.g. when e5 severely underestimates the number of wet days.
Therefore, $P$ is limited to a maximum of 1500 mm/day, which is close to the highest observed record at that time scale.

## 3.5    Consistency in time and space

To have consistent output in all version of HydroGFD3, there are internal checks to verify that each of the defined grid points
of HydroGFD3 is receiving data after each monthly adjustment. It happens that the land sea mask of the observational data sets
change over time, and they often differ between different data sources. If the anomaly data are not defined for a particular grid
point, a search algorithm will identify if there are defined anomalies in grid points within a 5 grid box radius. If the search is
successfully finding at least one value, the mean of all values in the search radius will be filling the grid point value. However,
if no defined data is found, the anomaly will be set to 0 for $T$ and 1 for $P$; in other words, the output will resort to adjustment
toward the HydroGFD3 climatology.

## 3.6    Evaluation

Evaluation of the HydroGFD3 historical data set is presented for the mean climatology of $P$ and $T$, as well as for regional
probability distribution functions (PDF) of daily data and as monthly mean time series. The two latter evaluations are performed
for each of the regions defined by Giorgi and Bi (2005) (although we use the correct longitude and latitude coordinates provided
by Huebener and Körper (2013)) commonly referred to as Giorgi regions, see Fig. 2. One exception is that we have left out the
EQF region in the plots of PFDs, since it is contained in other included regions. The reason is that it overlaps other regions,
and having only 25 regions simplifies the presentation layout of the plots substantially. For both the PDFs and the time series,



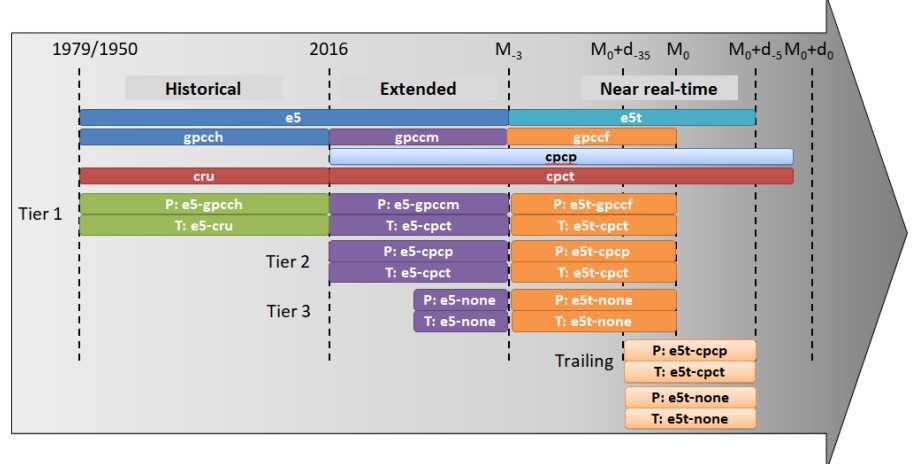

**Figure 3.** Schematic of the different HydroGFD3 products on a non-linear time axis. The top bars show the original data sources, and the Tier 1–3 and Trailing products are shown below. Abbreviations follow Tab. 1. The time axis denotes years with significant changes in data sources, and the later time marks are relative to the $1^{st}$ of the current month, $M_0$, and the current day, $d_0$. The units of the sub-script for the month is in months, and for the day is in days.

only data points in the defined grid points of HydroGFD3 are used. The PDFs are pooling all data in each domain, whereas the
time series plots are based on regional averages for each monthly time step.

## 4 Data sets

HydroGFD3 is built up by different data sets depending on the time period and the tier, see schematic in Fig. 3.
The historical period (1979–2016) is built on e5 corrected with the gpcch and cru data sets, respectively for for $P$ and $T$.
There is only one tier produced for this period. e5 will later be released back to 1950, and the HydroGFD3 historical data will
then cover that period as well.
After 2016, in the "extended" and "near real-time" periods, there are three tiers built on different data sets. Tier 1 is the
primary choice and follows the gpccm (for the e5 period) and gpccf (for the e5t period) products for $P$ adjustments, and the
cpct product (for the complete period) is used for $T$. Tier 2 builds instead on the cpcp and cpct products. Note that the Tier 1
and Tier 2 $T$ products are identical, and are only repeated here for simplification of the schematic. In practice, there is no Tier
2 for $T$, and the tiers are anyway not necessarily used consistently for $T$ and $P$ together, since the data sets are completely
independent. Tier 3 is the final resort if none of the data sets for a variable is available. It is performing only a climatological
correction of e5 or e5t by calculating anomalies of the reanalysis and adding/multiplying this to the HydroGFD3 climatology.
Since it does not make use of any observational data sets, it has received the internal file naming convention "none". For $P$, also
the number of wet days is adjusted, according to the description in Section 3.3, using the reanalysis anomalies as a predictor.



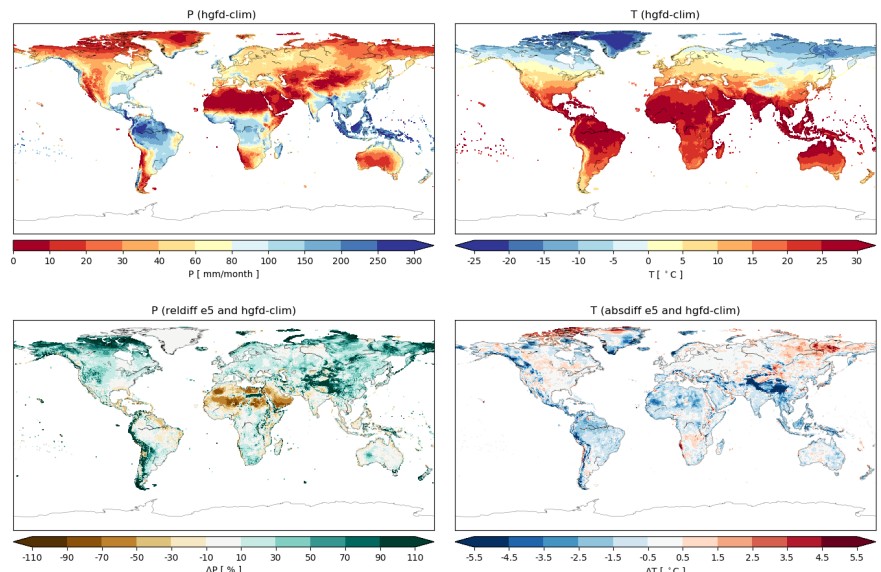

**Figure 4.** The baseline HydroGFD3 annual mean climatology for $P$ (top left) and $T$ (top right). The bottom row shows the bias of the e5 reanalysis for each variable.

A closer to real-time product is possible, with the daily time step cpcp and cpcpt products being available with a two day latency, and e5t available at five day latency. The adjustment of the e5t data is then based on the latest available 30 days, synchronized between the data sets, and is therefore called "Trailing".

## 4.1 Operational aspects

The HydroGFD3 data sets are updated at regular intervals. The "extended" period is updated each month, as new e5 and other data sets become available. Each tier works independently, and can therefore become available at different times.

The "near real-time" period is updated at earliest five days into the new month, when e5t is available. By then, the cpcp and cpcpt products are generally available, but gpccf normally needs a few days more. Tier 3 needs no additional data sets, and is available together with e5t, but is produced at the calendar month timestep like the other products. The priority order is independent for each variable, and goes from Tier 1–3.

Finally, the "Trailing" updates are performed along with e5t and cpcp and cpct updates, and is normally available at a five days time lag.

Earth System
Science
Data

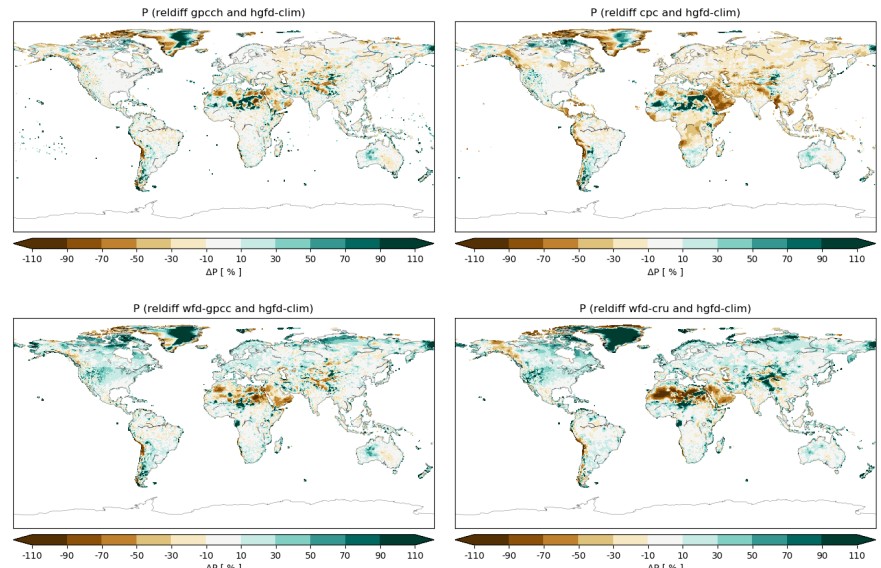

**Figure 5.** Relative difference of data sets to the HydroGFD3 annual mean $P$ climatology for the period 1980–2009; gpcch (top left), cpcp (top right), wfd-gpcc (bottom left) and wfd-cru (bottom right).

## 5  Results

### 5.1  Climatology

The climatological period of HydroGFD3 is set to 1980–2009, and is consistently used in this section. Figure 4 presents the annual mean climatology of HydroGFD for both $P$ and $T$, as well as the bias of the e5 reanalysis. e5 has in general a wet and cold bias in mountainous regions in most of the world. The Arctic is generally wetter and warmer in e5; note that Greenland $P$ is bias free per definition since the HydroGFD3 climatology uses e5 there. The tropics are generally drier and colder in e5.

Figures S1–S4 show the seasonal HydroGFD3 climatology and biases of e5. The bias patterns are rather stable across the seasons, although the magnitude changes somewhat. Most striking are the relative changes in western Africa in the December–February period, but this is the dry period there and the relative changes are therefore comparing low numbers which tend to exaggerate the absolute term differences.

We also compare the HydroGFD3 climatologies to other data sets, mainly with a focus on data with daily time steps that could be used equally for the historical period, but also to gpcch, which is the main background data set for anomalies in the historical period. Figure 5 shows the annual mean difference in $P$ of gpcch, cpcp, wfd-gpcc, and wfd-cru to the HydroGFD3 climatology. Differences to gpcch are generally within +/- 10 %, except for parts of the Andes mountain range, the Canadian Arctic, the dry north of Africa, the Himalayan plateau, and Greenland. The cpcp data set is generally drier, especially in the Arabian pensinsula. wfd-gpcc and wfd-cru are both generally wetter than the HydroGFD3 climatology, especially in the cold seasons (see Fig. S5–S8). This is due to the gauge corrections applied in the data, which is also the reason for wfd-gpcc not

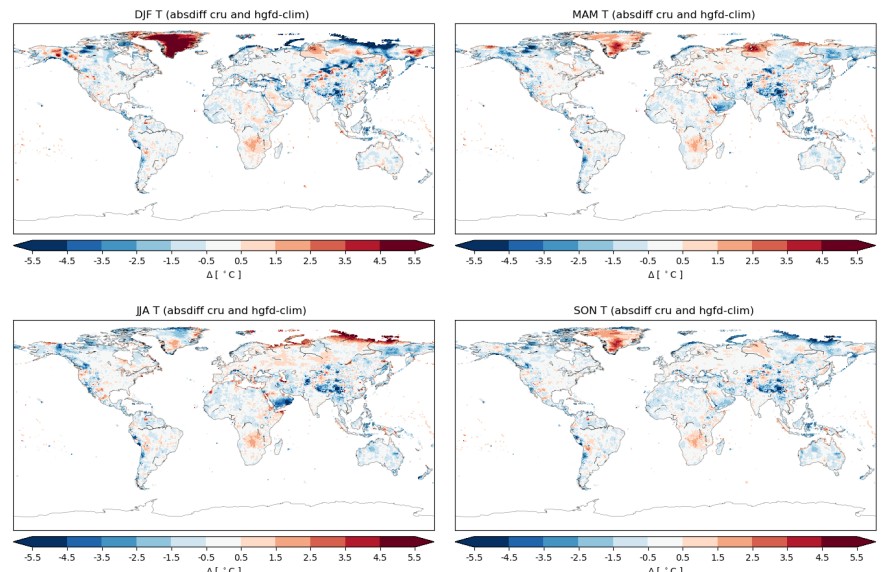

**Figure 6.** Absolute difference $T$ climatology for the period 1980–2009 between cru and HydroGFD3 for each season (top left) December–February, (top right) March–May, (bottom left) June–August, and (bottom right) September–November.

being identical to gpcch, which it is based on. However, the two wfd data sets also tend to be drier in very dry areas, which is likely due to the direct use of the number of rain days from the CRU data set. An incompatibility between $P$ and no observed wet days can act to remove $P$ completely for some months, and therefore making a drier data set. Seasonal differences (Fig. S5–S8) show similar patterns as the annual mean for most of the regions, but can also differ substantially in some regions. One region that stands out is southern Africa in JJA, where both gpcch and cpcp shows much wetter conditions (Fig. S7).

For $T$, we compare to cru only, since cpct is used to build the climatology, and wfd-cru is adjusted to cru and is per definition identical regarding climatology. Fig. 6 shows the absolute difference of cru and the HydroGFD3 climatology for each season of the climatological year. The largest differences are in the Arctic region where gauge availability is low. In other regions, such as central-south Africa, the Himalayan plateau, and other orographic regions, the differences are very consistent over all seasons, with deviations up to a few degrees Celsius. This makes us suspect that they are due to difference in the elevation used for the different data sets. The cpct data set does not come with any information on the elevations used. The use of anomalies from the cru and cpct in constructing the final data set removes such effects, but the climatological difference remains.

## 5.2 Distributions

Figure 7 shows the PDFs for the complete time period 1980–2009 for $P$, and for each of the data sets e5, hgfd3, cpcp, wfd-cru and wfd-gpcc. In these plots, the spread between the coloured lines representing direct observations or e5 adjusted to observations, can be interpreted as indicators of the uncertainty in the observed state. Many regions show fairly high agreement between the datasets, including the e5 original data. In some regions, there is a large spread in the observations, and e5 is





somewhere in between, e.g. in ALA, GRL, TIB, and SAH. However, in other regions e5 is deviating significantly in part of the
distribution, such as in SSA and WAF moderate intensities, AMZ and EAF extreme intensities.
HydroGFD3 tends to have higher extremes than other datasets. This is partly a resolution effect due to the 0.25 degree
resolution of HydroGFD, and 0.5 degree of the other data sets used here. A coarser resolution will move all higher intensities
toward the lower intensities (to the left in the PDF plots). That the effect differs between regions is because the extremes are
also modulated by the magnitude of the applied correction, i.e. the applied scaling. A scaling factor above one will increase
the extremes, and below one will decrease them. The baseline climatology therefore has an impact on the extremes. Also the
wet-days calculation of HydroGFD3 can affect the results, and we find that the dry regions, e.g. SAH and MED, has more
dry days in HydroGFD3 than in the other data sets. When e5 only gives few $P$ days, while the observational anomaly is
high, the scaling factor can become very large, and the only process to limit this is the upper limit of 1500 mm/day, which is
seldom reached. The wfd-gpcc, which has a similar methodology as HydroGFD3, still has lower extremes. Besides the above
mentioned under-catch corrections, the lower extremes may be due to the upper threshold applied to each hour, as can be seen
in the original wfd-code in the CDS-catalogue (https://doi.org/10.24381/cds.20d54e34).
For $T$, the general shapes of the PDFs agree across all data sets and regions. However, there are sometimes substantial
differences between e5 and the observational data sets. Typically, e5 displays issues around 0 °C, which is common in global
models and related to melting conditions. There are also seasonal offsets outside the range of the observations. HydroGFD3
remains fairly close to cpct and wfd-cru in most cases. Orographic effects on the $T$ was not accounted for in this comparison,
which can explain some of the differences in regions with varying orography such as TIB.

## 5.3 Temporal trends

To get an impression of the temporal trends, and to identify potential issues in the time series, we also investigate the time series
as an average over the Giorgi regions. To emphasize differences between the data sets, we discuss mainly differences relative
to a common reference, here chosen to be e5. In other words, we present the inverse bias of e5 compared to each observational
source.
Figure 9 shows the results for $P$ for the period 1980–2019, and the absolute values are shown in Fig. S9. Note that wfd-gpcc
ends in 2016, wfd-cru ends in 2018, and gpccm and gpccfg are only available for the last years. The most striking feature is
the strong deviations of cpcp for many of the regions. It also varies significantly with time by changing variance, e.g. in SEA,
changing mean value, e.g. in CAS, SAS, and AMZ. In some years, there are significant offsets compared to surrounding years,
e.g. in 2014 in NEU, NEE, CAS and MED. Likely, these issues are due to variations in the underlying station network, but we
have not verified this. wfd-gpcc and wfd-cru display stronger anomalies over the annual cycle in the colder regions compared to
other data sets. This is likely due to the undercatch corrections which are larger for snowy conditions. As expected, HydroGFD3
follows the general trends of wfd-gpcc, and the other data sets have similar trends, besides the cpcp deviations just discussed.
The gpccm and gpccf has similar mean and variance as gpcch in the overlapping period, and shows generally consistent
behavior for the later years. Although, some larger anomalies occur in, e.g., CAN, CAM, SQF, and SAH.

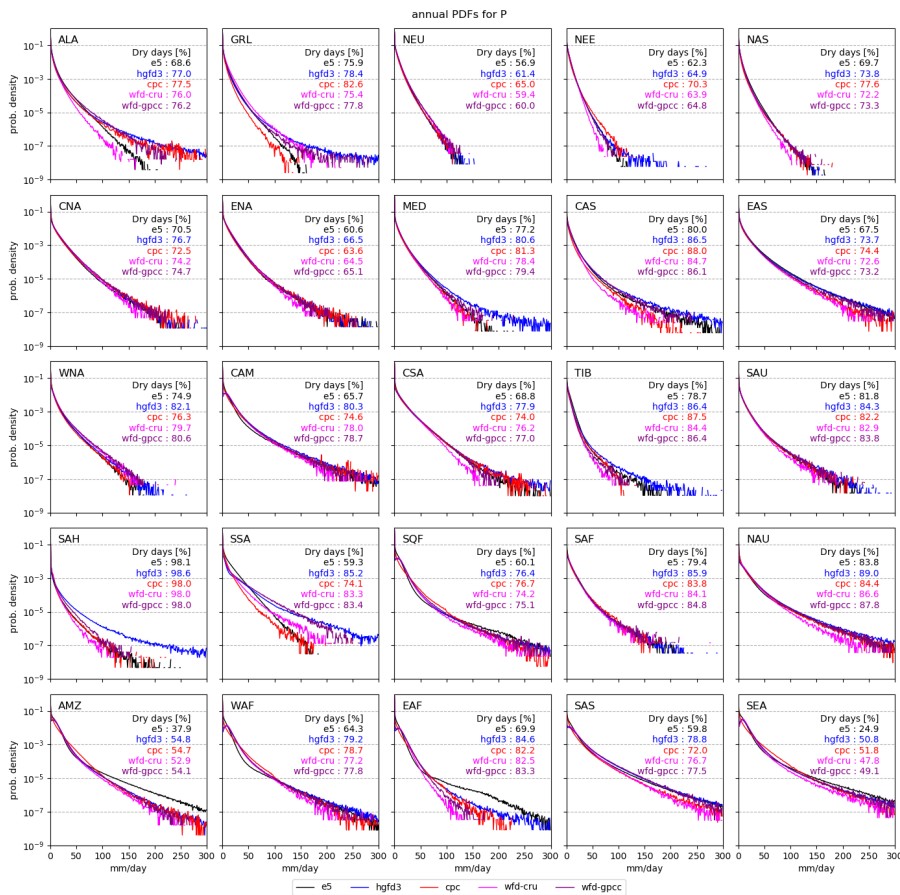

**Figure 7.** $P$ PDFs of each Giorgi region for the data sets with daily output data in the period 1980–2009. The table in each plot states the percentage of dry days for each data set, i.e. the percentage of data in the first bin of 0–1 mm/day.

For $T$, the anomalies to e5, see Fig. 10 (and Fig. S10), retain a clear annual cycle in many regions. Sometimes, the annual
cycle is mainly for wfd-cru (e.g, NEU, TIB, SAS), but often for all data sets. HydroGFD3 and cpct are in general close to each
other, because of the HydroGFD3 climatology reducing the offset to zero. However, cpct has some clear "break points" in its
time series in some regions. For example, in NEU, there is a marked change in the magnitude of the anomalies from about 0
to 0.5 °C to -0.5 to 0.5 °C about year 2006. A similar change about that time is visible also for EAS, GRL, MED, SAS, and
NAU. Because the climatologies are calculated for the period 1980–2009, part of these changes are included with the earlier
weaker variability. HydroGFD3 is based on cru anomalies pre-2017, but from 2017 on, also its variability is subjected to the
changes in cpct.
Some regions display a significant offest between the data sets, such as SEA, CSA, MED, TIB, and SAS, with cru having
generally lower $T$s. Interestingly, changes in cpct after 2006 often act to reduce the offset to cru.



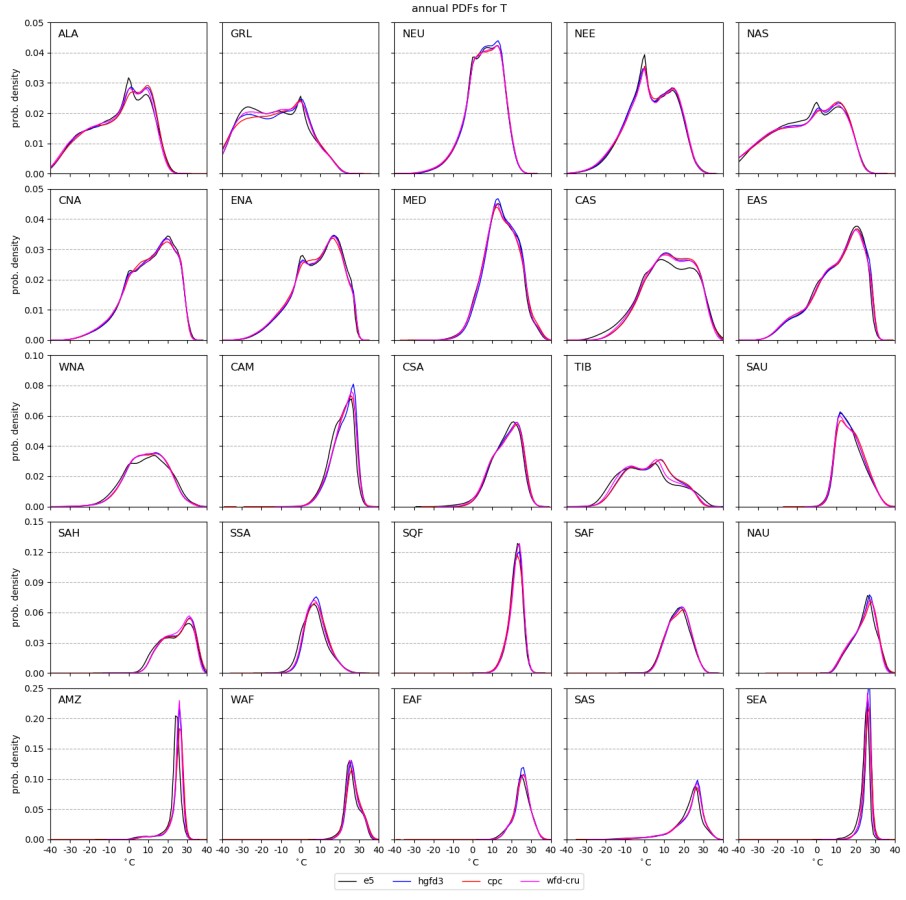

**Figure 8.** *T* PDFs of each Giorgi region for the data sets with daily output data in the period 1980–2009.

## 5.4 Extending to near real-time

The near real-time products, in Fig. 3 called "trailing", use the daily updates of the cpcp and cpct observations. They are
therefore subject to the quality of the cpc products, and the changes in time as discussed in the previous section. This product
follows HydroGFD3 fairly closely to that shown in Fig. 9 and 10, as the main version Tier 2 is also based on cpcp and cpct,
but with corrections at calendar months.
In addition, also the "none" products are created with the trailing time window. These only replace the e5 climatology with
that of HydroGFD3, and is the simplest form of corrections of the mean. They act as the last failover option in the production
chain, before defaulting to un-corrected e5 data. We do not present this product in the time series plots, since it would only
constitute a constant annual cycle offset in comparison to e5.

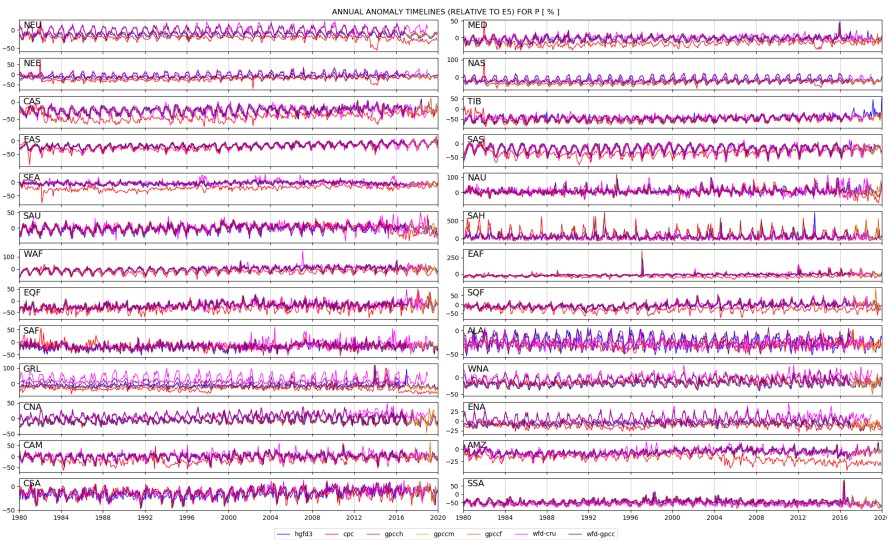

**Figure 9.** $P$ anomalies for all data sets, averaged over the Giorgi regions for all valid land data points. The anomalies are relative to the e5 data set, and is evaluated for each single month.

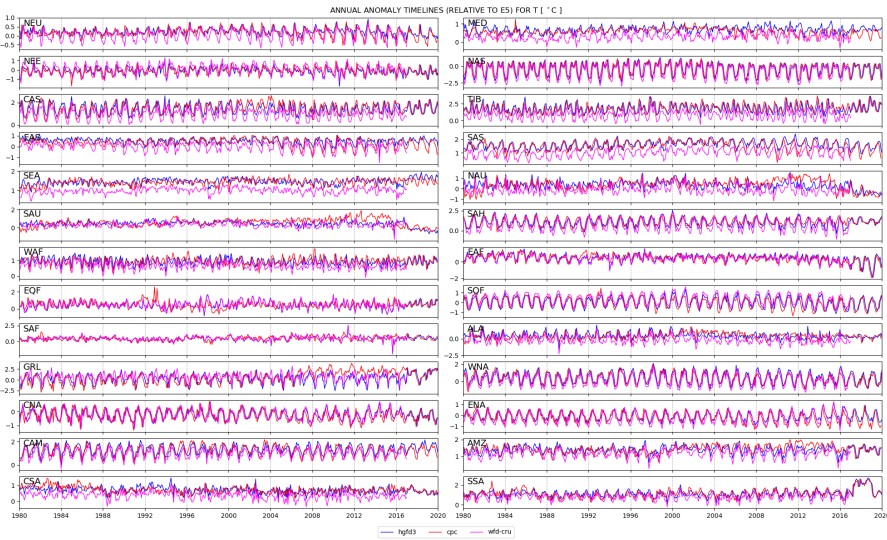

**Figure 10.** $T$ anomalies for all data sets, averaged over the Giorgi regions for all valid land data points. The anomalies are relative to the e5 data set, and is evaluated for each single month.

## 255  6   Discussion

Compared to similar data sets based on reanalysis, such as WFD and MSWEP, HydroGFD3 differs in that it has its own
climatological background, and performs the corrections based on anomalies of that same climatological time period. The





reason for using this method, as to be able to switch datasets closer to real-time, without "jumps" in the time series. This works well as long as the real-time data set retains its climatological state, which seems to be the case for gpccm and gpccf compared to gpcch. However, cpct and cpcp both cause issues due to changes in the time series towards the end of the time period, about year 2006. The bias of e5 is still reduced, which brings validity to the method.

HydroGFD3 has generally higher extremes than the other analyzed data sets. This is especially so in drier regions where an interplay between the estimation of the number of wet days and the scaling causes fewer wet days and larger scaling factors. In effect, this leads to enlarging the tail of the distribution. It is possible to restrict the scaling by only allowing the scaling factor to be a few times the original value, but such restrictions would in turn impact on the monthly mean. A potential method would be to "borrow" $P$ from adjacent grid points on e5's excessive dry days, and thereby reducing the scaling factors. This topic is being investigated for future updates of the methodology.

The regional analysis shows clearly that the observational data sets give substantially different results in some regions. Diverse results are more common in data sparse regions or in regions where data are not generally available to all data sets. It is therefore difficult to determine which is closer to the truth in a global assessment like this, and more detailed regional studies, such as Fallah et al. (2020), are needed.

The current main usage of the data set is to initialize different HYPE forecasting models around the world, e.g. in Europe (Hundecha et al., 2016), the Niger river (Andersson et al., 2017), and world-wide (Arheimer et al., 2020). This has influenced some of the choices for the setup, such as the use of only the ERA5 reanalysis model, among other reanalysis systems used in e.g. the MSWEP data set (Beck et al., 2017). The forecasts produced by these hydrological models are primarily using the ECMWF deterministic medium range forecasts, or the probabilistic SEAS5 seasonal forecasts, which both use the same model as e5. The priority order of the different redundancy options, i.e. the Tiers 1–3, is based on experience with using the different data sources for our forecasts, with impact from both availability for a given month as well as experienced longer interruptions.

## 7  Conclusions

The HydroGFD3 methodology of correcting the e5 reanalysis model toward an observational reference, along with the resulting data sets were presented. We conclude that the data sets compare well with existing similar data sets.

The main new features of HydroGFD3 are:

– Higher spatial resolution of 0.25 degrees.

– Near real-time corrected data until five days from now, i.e. following the continuous updates e5 + e5t time period.

– Temporal coverage from 1979, and will be extended back to 1950 along with the extended e5 data expected during 2020.

– Multiple redundancy options to avoid halting production when single data sets are delayed.

The data is freely available for the period 1979–2019, and by subscription for the real-time products. See Section 8 for details.





## 8 Data availability

A historical period, ranging from February 1979 to December 2019, is available as open source from the ZENODO repository at doi:10.5281/zenodo.3871707. For years prior to 2017, cru and gpccm are used as reference data for $T$ and $P$, respectively. The following years use instead gpccm and cpct reference data.

Real-time updates of the data set are available for a processing charge via subscriptions. Please make a request here: https://hypeweb.smhi.se/buy-water-services/data-subscription/ and make sure to mention the data set name "HydroGFD3".

*Acknowledgements.* The authors would like to thank the EU and the Swedish Research Council Formas for funding, in the frame of the GlobalHydroPressure project financed under the 2017 Joint Call of Water JPI (IC4WATER).



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
