# Peer review of "HydroGFD3.0: a 25 km global near real-time updated precipitation and temperature data set"

_Earth System Science Data, 2020_

## Referee Comment (RC1) · Graham Weedon (Referee) · 20 Oct 2020

Overview:

Berg et al. describe the processing procedures needed to generate the HydroGFD3 data set from the ERA5 reanalysis. The data set covers hourly precipitation rate, near-surface air temperature and daily mean, maximum and minimum near-surface air temperature at 0.25o x 0.25o spatial resolution. The freely available data cover February 1979 to December 2019, but later, near real-time data (up to 5 days ago) requires a subscription.

The processing methodology builds on the WATCH Forcing Data (WFD) procedures (Weedon et al., 2011, JHM, Weedon et al., 2014, WRR) that were used in HydroGFD1

(Berg et al., 2018, HESS). The paper compares HydroGFD3 with the precipitation and temperature components of the WFDE5 data (Cucchi et al., 2020 ESSD, though these data are at 0.5o x 0.5o resolution) and with the observational datasets used in the processing to establish the climatologies needed for bias correction.

The descriptions of the processing procedures are clear - though I may well be biased since I designed the WFD methodology on which they are largely based. Comparisons of HydroGFD3 with WFDE5 and the contributing data sets are favourable aside from a problem with the probability of rainfall in the MED and SAH regions (Fig. 7). In these areas the authors have noted/acknowledged (lines 262-264) that they have far higher probabilities at almost all rainfall rates than the comparison data sets probably due to the reduced numbers of wet days used.

Aside from some minor issues and text corrections, there is one key weakness with the manuscript which I will raise below. Otherwise the manuscript is well presented and close to acceptable for publication.

Key weakness:

The key weakness concerns the evaluation. As explained HydroGFD3 derives from HydroGFD1 which was based on the WFD methodology. Consequently, it seems likely that HydroGFD3 will inevitably compare favourably with the WFDE5 data set since the latter was also produced using the WFD methodology (Cucchi et al., 2020, ESSD). Similarly, the various data sets listed in Table 1 have to an extent been incorporated into HydroGFD3 so again comparison with the new data set is liable to be favourable. My recommendation therefore is that the authors re-evaluate HydroGFD3 using truly independent data set (for example, using global data produced using a non-WFD methodology and without using an ERA reanalysis and/or using site observations e.g. FLUXNET2015 data).

Nomenclature issue:

Line 63: "WATCH forcing data WFDE5" is NOT correct and is misleading. WFDE5 means the "WATCH Forcing Data methodology applied to ERA5 reanalysis" (Cucchi et al., 2020, ESSD). WFDE5 has nothing to do with the WATCH programme (which ended in 2011). The authors should avoid using "wfd" as a shorthand for WFDE5. Correct all uses of "wfd" to WFDE5 or wfde5 throughout the manuscript (i.e. Table 1, 181, 185, 186, 191, 199-200, 213, 219, 226-227, 231, 233, 256, Figs 5, 7, S5, S6, S7, S8).

Update:

Cucchi et al. (2020 ESSD) has been revised and is now fully published – update the reference list.

Issue of interpretation:

My issue of interpretation concerns lines 186-188: "However, the two wfd [WFDE5] datasets also tend to be drier in very dry areas, which is likely to due to the direct use of the number of rain days from the CRU data set. An incompatibility between P and no observed wet days can act to remove P completely for some months, and therefore making a drier data set." This is the wrong explanation for why WFDE5 data appear drier than HydroGFD3 in Fig. 5. Firstly, the WFDE5 monthly precipitation totals are adjusted after the correction of the number of wet days. So, as long as there are wet days, the wet day correction does not lead to too little monthly precipitation.

Secondly, suppose for a grid box the number of ERA5 wet days were to be set to zero from using CRU wet days even if there is some precipitation according to CRU or GPCC. This situation would mean that all the days with ERA5 precipitation would have been reset to zero in the WFDE5 processing. However, the adjustment of a hourly ERA5 rate involves multiplication by the CRU or GPCC monthly total divided by the (supposedly zero) monthly ERA5 precipitation total. This would would generate a "NaN". Such as result would have caused the WFDE5 processing to stop since there is a check for plausible precipitation rates for every grid box and every time step. Hence, there are never occasions in the WFDE5 processing when the CRU number of wet

days is zero but the ERA5 and CRU totals exceed zero. As a result, there needs to be a different reason given for the observation that the WFDE5 data are drier than HydroGFD3 in the dry regions in Fig. 5.

Minor text changes (L = line number):

L26 Add Cucchi et al 2020

L74 Used here "conservatively" sounds like it means being cautious rather than ensuring conservation of total precipitation during interpolation/remapping. Reword the sentence.

L77 boarders > borders

L94 1989-2009 shouldn't this be 1980-2009 as used everywhere else?

L127 version > versions

L131 successfully > successful in

L131 will be filling > is used to fill

L132 data is > data are OR data is > data value is

L139 PFDs > PDFs

L185 "the data" is ambiguous change to "wfde5 data"

Fig 7 cpc > cpcp

Fig 8 cpc > cpct

Fig 9 cpc > cpcp

Fig 10 cpc > cpct

L252 failover > failsafe [wrong use of failover]

L258 as to > is to

L264 enlarging the tail of the distribution. > enlarging the tail of the distribution (e.g. in the MED and SAH region in Fig. 7).

---

## Referee Comment (RC2) · Anonymous Referee #2 · 23 Nov 2020

General comments:

The authors present a new version of the HydroGFD data set, which contains (nearly) global daily precipitation and temperature reanalysis data since 1979 at 0.25° resolution. This data set is based on the ERA5 reanalysis data from the ECMWF, which are corrected on the basis of several observation-based gridded data sets. HydroGFD3.0 is meant to be used e.g. as atmospheric forcing data set for hydrological modelling and for impact studies.

This kind of bias adjusted data sets is certainly very useful, in particular since i) it has a global coverage (except Antarctica), and ii) it will be prolonged in near real time, making it suitable for operational applications.

Besides some ambiguities (see below), the paper is pleasant to read. However, I have two major remarks about the methodology:

1) About the evaluation The authors mainly compare the HydroGFD3.0 data set with the observational data sets they use to calculate the bias adjustment and with the WFDE5 data set, which also relies on these same observational data sets. Thus, I cannot completely agree with the author's conclusion on I. 281. Comparing HydroGFD3.0 to the data sets it relies on (observation-based data sets and ERA5) is very interesting as it evaluates the impact and the usefulness of the bias correction applied to the ERA5 data. However, the consistency and reliability of HydroGFD3.0 could be shown in a much more convincing way, if the authors could add an in-depth evaluation by comparing HydroGFD3.0 with at least one independent data set. Further, as HydroGFD3.0 is a new version of already existing data sets (HydroGFD1.0 and HydroGFD2.0), it might be interesting to add a comparison with these previous versions. This would show the improvements and the added value of v3.0 compared to the previous versions and thus the interest of using this new data set.

2) About the long term changes in the input data sets To my understanding, any changes in the variability within or between data sets used for the bias adjustment, but also long term trends within these data sets will be transposed to HydroGFD3.0. As both the variability and long term trends might be different from one data set to another, the consecutive use of different data sets over time might induce discontinuities in the HydroGFD3.0 data. This aspect, along with a short comparison of the variability and the long term trends between consecutive data sets, should be discussed in the manuscript.

Specific comments:

Section 2 : It would be very useful for the understanding of the paper, if the authors could give more details about the data sets they use, e.g. better differentiate if a data set is a reanalysis or interpolated observations, the specificities of each data set and
especially in which way it is complementary to or different from the other data sets. The authors should also expand the acronyms (CRU, CPC, etc.) the first time they are used.

Section 2 : The authors talk about background climatology (I. 56), historical period (I. 57, which period ?), climatological adjustment (I. 60, 61), etc. which are only explained later in sections 3 and 4. It might be easier to follow if section 2 only described the data sets used (see comment above), and if paragraph I. 56 was moved at a more appropriate place in section 3.

I. 53 : "with a similar model as that used for ERA5". It would be useful to add some more information here (Is it the same model ? Are the biases supposed to be similar ?).

Section 3 : It would be interesting if the authors could add some explanation on why they calculate the correction on a monthly basis and not e.g. on a 30-day running mean basis around each day, which would smooth the correction curve applied to ERA5.

Section 3 : As the authors use monthly, daily, and hourly data, and also process the data on monthly, daily, and hourly time scales, it would be very useful for understanding to always clearly mention the time scale e.g. on which corrections are applied, anomalies are calculated, especially in this section, but also throughout the whole manuscript. For example, it is not clear to me whether the monthly corrections are applied to hourly or daily P.

I. 75: How / in comparison to which reference were the issues in CHPclim identified ?

I. 80 and 82 : Is the remapping of T and Nwet also conservative as for P?

I. 83 : "retaining the grid points that are available consistently in all data sets and all months." Does this mean that HydroGFD3.0 data are not available for some land grid points ?

I. 102 : The dry and wet days should be defined clearly here at latest. Is the threshold

ESSDD
of 1 mm/day mentioned in the caption of Fig. 7 used to separate dry from wet days ?

I. 106 : Does N\_clim\_wet come from the CRU data set ?

Section 3.4 : This synthesis of the correction steps is very useful. However, it might be a little bit more detailed : - To my understanding, the very first step is the preparation of the climatology. - Step 1 : "Calculate monthly anomalies in observation data"? - Step 4 : the removal of the weakest excessive wet days should be (more clearly) explained in section 3.3. - Steps 5 and 6: This is not clear to me. Does it only concern P? If not, I do not understand what the ratio stands for.

I. 182-183 : It might be interesting to add some more discussion, e.g. remember as said before that the relative biases for P are much higher in these dry (or snowy) regions. Moreover, as no correction is applied over Greenland and the biases are quite huge, should one conclude that the climatology is not reliable there ? It might be useful to briefly discuss the interest of not excluding Greenland as it is done for Antarctica.

I. 191 : "For T, we compare to cru only, since cpct is used to build the climatology": - This is confusing as following e.g. section 4 (e.g. I. 145), fig. 3, and I. 291, cru seems to be used for the climatology, but I. 56 mentions cpct. - In any case, it might be interesting to compare with both data sets to see the impact of the bias adjustment, as it is done for P with gpcch.

I. 203-204 : It would be interesting to add some explanation on why the biases are larger in these regions (see also comment for I. 182-183).

I. 210 : Correct "... dry regions ... have more dry days ...". Are the differences really significant / worse to mention ? The values seem to be very similar.

Paragraph I. 226 : It seems that many regions show an annual cycle. This should be briefly discussed, as it is done for T.

I. 283 : Which was the resolution of the previous versions ?

**ESSDD**
I. 286 : I agree and this is a major advantage of this data set. However, would such a switch between data sets not introduce biases (see e.g. the general comment on the variability and long term trends) ?

I. 287-288 : As section 8 follows directly, it is redundant to already mention this here.

Technical corrections:

I. 8 : Correct "... as well as the number of wet days ..."

I. 49 : "tiers" are explained in section 4 "Data sets" not in the methodology. It might be useful to already explain here in a few words what is meant by "tiers".

I. 52 : Expand "SMHI".

- Tab. 1 : Nwet acronym is not defined.
- I. 71 : "climatological period" Is this 1980-2009 ?
- Section 3.1 : Use data set names of Table 1 (e.g. CHPclim, GPCCv8).
- I. 94 : Correct "1989-2009" to "1980-2009".
- I. 128 : Correct "It happens that the land sea masks ...."
- I. 132 : Correct "... the output will resort no adjustment ...."?
- I. 145 : Correct "... respectively for P and T."

Fig. 4 and I. 172 : With respect to which data is the bias of e5 shown here ?

Fig. 4 : It might be more consistent to represent Greenland in another way, e.g. like Antarctica, as no bias could be calculated.

- I. 157 : Correct "... cpcp and cpct products ...."
- I. 164 : Correct "... cpcp and cpct products ...."
- I. 172, 206 : Correct "HydroGFD3"
- I. 184 : Correct "Arabian peninsula."
- I. 190 : Correct "...where both gpcch and cpcp show ...."
- I. 195 : Correct "... they are due to differences in elevation ...."
- I. 216 : Add reference to Figure 8.
- I. 219 : Correct "Orographic effects on T were ...."
- I. 234 : Correct "... have similar mean ... and show generally ...."
- I. 242 : Should it not be 2016 instead of 2017 ? If not, Fig. 3 should be adapted.
- I. 244 : Correct "... a significant offset between the data sets ...."
- I. 245 : "... reduce the offset to cru." Should it not be to e5 ?
- Fig. 9 and 10 : Are these monthly anomalies ? Correct "... and are evaluated ...."
- I. 258 : Correct "... using this method is to be able ...." ?
- I. 292 : Switch gpccm and cpct to be consistent with the previous sentence.

---

## Editor Comment (EC1) · Martin Schultz (Editor) · 28 Dec 2020

Dear authors, when preparing your manuscript revision, please consider adding some of the discussion about independent datasets which you provided in the answers to both reviewers to section 3.6. Thank you!

---

## Author Response (AR1)

**Dear editor,**

**Here follow the previously given answers to reviewers, with *added information* on how we have addressed the comments in the revised manuscript.**

**Peter Berg, on behalf of all authors**

**Reviewer no. 1**

**Dear Graham,**

**Thank you very much for the detailed reading and the useful comments to our manuscript. Below, we answer in bold font to all issues raised.**

Overview:

Berg et al. describe the processing procedures needed to generate the HydroGFD3 data set from the ERA5 reanalysis. The data set covers hourly precipitation rate, near surface air temperature and daily mean, maximum and minimum near-surface air temperature at 0.25o x 0.25o spatial resolution. The freely available data cover February 1979 to December 2019, but later, near real-time data (up to 5 days ago) requires a subscription.

**A minor correction is in place here. The data set does not include the hourly precipitation rates, only aggregations to the daily scale. It is indeed possible to construct data at this frequency with the methodology, as you also do in Cucchi et al. (2020), we have chosen not to make this data available until we have analysed this more thoroughly. The hourly time scale is often poorly represented in coarse model data, and data for verification is even more difficult to get hold of.**

The processing methodology builds on the WATCH Forcing Data (WFD) procedures (Weedon et al., 2011, JHM, Weedon et al., 2014, WRR) that were used in HydroGFD1 (Berg et al., 2018, HESS). The paper compares HydroGFD3 with the precipitation and temperature components of the WFDE5 data (Cucchi et al., 2020 ESSD, though these data are at 0.5o x 0.5o resolution) and with the observational datasets used in the processing to establish the climatologies needed for bias correction.

The descriptions of the processing procedures are clear - though I may well be biased since I designed the WFD methodology on which they are largely based. Comparisons of HydroGFD3 with WFDE5 and the contributing data sets are favourable aside from a problem with the probability of rainfall in the MED and SAH regions (Fig. 7). In these areas the authors have noted/acknowledged (lines 262-264) that they have far higher probabilities at almost all rainfall rates than the comparison data sets probably due to the reduced numbers of wet days used.

Aside from some minor issues and text corrections, there is one key weakness with the manuscript which I will raise below. Otherwise the manuscript is well presented and close to acceptable for publication.

Key weakness:

The key weakness concerns the evaluation. As explained HydroGFD3 derives from HydroGFD1 which was based on the WFD methodology. Consequently, it seems likely that HydroGFD3 will inevitably compare favourably with the WFDE5 data set since the latter was also produced using the WFD methodology (Cucchi et al., 2020, ESSD). Similarly, the various data sets listed in Table 1 have to an extent been incorporated into HydroGFD3 so again comparison with the new data set is liable to be favourable. My recommendation therefore is that the authors re-evaluate HydroGFD3 using truly independent data set (for example, using global data produced using a nonWFD methodology and without using an ERA reanalysis and/or using site observations e.g. FLUXNET2015 data).

**We agree that a complete evaluation should use independent data. Truly independent data are rare at a global scale since most large data sets are already included in the gridded observations. However, our aim is to provide a comprehensive overview of the data to present its qualities and also to point to potential issues. To this purpose, we compare to other data sets at a global scale, with specific analyses for sub-regions. The analysis is based on a range of data sets that have been completely or to a large extent independently generated.**

**For precipitation, the HydroGFD3 climatology is based on the satellite-gauge merged CHPclim climatology, with some minor additions of GPCCv8 for Scandinavia and part of Siberia. Therefore CPC-Unified, GPCCv8, wfd-gpcc and wfd-cru can all be considered independent data sets and are valid for the evaluation in Fig. 5. For the daily distribution and timeseries in Figs. 7 and 9, HydroGFD3 is based on climatology with GPCCv8 anomalies, and is therefore independent of CPC-Unified, partly independent of WFDE5-CRU (besides ERA5), and to some extent independent of WFDE5-GPCC regarding the climatology.**

**For temperature, the HydroGFD3 climatology is based on CPC-Temp, and CRU is therefore independent data. The daily HydroGFD3 version for temperature is based on climatology and CRU, and is indeed strongly dependent of CPC, and partly independent of WFDE5-CRU (besides ERA5). Please note that another reviewer spotted an inconsistent use of CRU and CPC-Temp data in the manuscript, which will be amended for the revision.**

**The daily timeseries are plotted as a visual guide to spot issues with trends of the different data sets, with specific focus on the last years when the different tiers might affect the temporal consistency. It is not a full-fledged evaluation, but is there to give an impression of the quality of the product. Our experience from earlier studies is that in-depth evaluation can only be performed at the local scale, e.g. Fallah et al., (2020; in the reference list), and we will encourage users to pursue such evaluations.**

*We have addressed this by adding a note about the independence between data to raise this issue to the attention of the reader:*

"An issue with global scale evaluations is that of independence between data sets, and most of the gauge-based data sets listed in Tab. 1 make use of more or less the same openly available observations, with regional differences. The data sets have, however, been independently generated and use different statistical models for the gridding process. Our aim is to provide a comprehensive overview of HydroGFD3 in comparison to other data sets, in order to presents its qualities and to point out potential issues. For each of the comparisons in Section 5, we chose data sets that are as

independent as possible, given the limitations just discussed. Our experience from earlier studies is that in-depth evaluation can only be performed at the local scale (e.g. Fallah et al. (2020)), and we encourage users of the data set to pursue such evaluations.*"*

Nomenclature issue:

Line 63: "WATCH forcing data WFDE5" is NOT correct and is misleading. WFDE5 means the "WATCH Forcing Data methodology applied to ERA5 reanalysis" (Cucchi et al., 2020, ESSD). WFDE5 has nothing to do with the WATCH programme (which ended in 2011). The authors should avoid using "wfd" as a shorthand for WFDE5. Correct all uses of "wfd" to WFDE5 or wfde5 throughout the manuscript (i.e. Table 1, 181, 185, 186, 191, 199-200, 213, 219, 226-227, 231, 233, 256, Figs 5, 7, S5, S6, S7, S8).

**Thank you for the clarification.**

***We have adopted your suggestion and rename it in the revision.***

Update: Cucchi et al. (2020 ESSD) has been revised and is now fully published – update the reference list.

**Thank you, we will of course update this reference.**

Issue of interpretation:

My issue of interpretation concerns lines 186-188: "However, the two wfd [WFDE5] datasets also tend to be drier in very dry areas, which is likely to due to the direct use of the number of rain days from the CRU data set. An incompatibility between P and no observed wet days can act to remove P completely for some months, and therefore making a drier data set." This is the wrong explanation for why WFDE5 data appear drier than HydroGFD3 in Fig. 5. Firstly, the WFDE5 monthly precipitation totals are adjusted after the correction of the number of wet days. So, as long as there are wet days, the wet day correction does not lead to too little monthly precipitation.

Secondly, suppose for a grid box the number of ERA5 wet days were to be set to zero from using CRU wet days even if there is some precipitation according to CRU or GPCC. This situation would mean that all the days with ERA5 precipitation would have been reset to zero in the WFDE5 processing. However, the adjustment of a hourly ERA5 rate involves multiplication by the CRU or GPCC monthly total divided by the (supposedly zero) monthly ERA5 precipitation total. This would would generate a "NaN". Such as result would have caused the WFDE5 processing to stop since there is a check for plausible precipitation rates for every grid box and every time step. Hence, there are never occasions in the WFDE5 processing when the CRU number of wet days is zero but the ERA5 and CRU totals exceed zero. As a result, there needs to be a different reason given for the observation that the WFDE5 data are drier than HydroGFD3 in the dry regions in Fig. 5.

**Thank you for providing more insight on the WFDE5 processing on this issue. We agree completely in what you are stating. However, not knowing exactly how the situation of division by zero was programmed, we speculated about the reason. We can now conclude that the reason for the differences actually lies in a combination of the background climatology mixed with GPCC and that**

**zero-division is avoided in HydroGFD3 by setting the scaling to zero when this occurs. The manuscript will be updated accordingly.**

*We have updated the description:*

"There are also discrepancies in large dry desert areas, such as the Sahara desert, which arise due to differences in the way the number of wet days are calculated in the different data sets. The WFDE5 implementation would produce NaN in division by zero if the number of wet days was zero, which has not happened so far (reviewer comment by Graham Weedon). In HydroGFD3, division by zero does occur, and is solved by setting the ratio to zero when the calculated number of dry days equals zero. An incompatibility between P and no observed wet days can act to remove P completely for some months, and therefore making a drier data set."

Minor text changes (L = line number):

**We appreciate the detailed checks, and have adjust all in the revised version.**

L26 Add Cucchi et al 2020

L74 Used here "conservatively" sounds like it means being cautious rather than ensuring conservation of total precipitation during interpolation/remapping. Reword the sentence.

L77 boarders > borders

L94 1989-2009 shouldn't this be 1980-2009 as used everywhere else?

L127 version > versions

L131 successfully > successful in

L131 will be filling > is used to fill

L132 data is > data are OR data is > data value is

L139 PFDs > PDFs

L185 "the data" is ambiguous change to "wfde5 data"

Fig 7 cpc > cpcp

Fig 8 cpc > cpct

Fig 9 cpc > cpcp

Fig 10 cpc > cpct

L252 failover > failsafe [wrong use of failover]

L258 as to > is to

L264 enlarging the tail of the distribution. > enlarging the tail of the distribution (e.g. in the MED and SAH region in Fig. 7).

**Reviewer no. 2**

**Thank you for the thorough review of our manuscript, and for the very helpful comments on improvements.**

General comments:

The authors present a new version of the HydroGFD data set, which contains (nearly) global daily precipitation and temperature reanalysis data since 1979 at 0.25◦ resolution. This data set is based on the ERA5 reanalysis data from the ECMWF, which are corrected on the basis of several observation-based gridded data sets. HydroGFD3.0 is meant to be used e.g. as atmospheric forcing data set for hydrological modelling and for impact studies.

This kind of bias adjusted data sets is certainly very useful, in particular since i) it has a global coverage (except Antarctica), and ii) it will be prolonged in near real time, making it suitable for operational applications.

Besides some ambiguities (see below), the paper is pleasant to read. However, I have two major remarks about the methodology:

1) About the evaluation The authors mainly compare the HydroGFD3.0 data set with the observational data sets they use to calculate the bias adjustment and with the WFDE5 data set, which also relies on these same observational data sets. Thus, I cannot completely agree with the author's conclusion on l. 281. Comparing HydroGFD3.0 to the data sets it relies on (observation-based data sets and ERA5) is very interesting as it evaluates the impact and the usefulness of the bias correction applied to the ERA5 data. However, the consistency and reliability of HydroGFD3.0 could be shown in a much more convincing way, if the authors could add an in-depth evaluation by comparing HydroGFD3.0 with at least one independent data set. Further, as HydroGFD3.0 is a new version of already existing data sets (HydroGFD1.0 and HydroGFD2.0), it might be interesting to add a comparison with these previous versions. This would show the improvements and the added value of v3.0 compared to the previous versions and thus the interest of using this new data set.

**We agree that a complete evaluation should use independent data. Truly independent data are rare at a global scale since most large data sets are already included in the gridded observations. However, our aim is to provide a comprehensive overview of the data to present its qualities and also to point to potential issues. To this purpose, we compare to other data sets at a global scale, with specific analyses for sub-regions. The analysis is based on a range of data sets that have been completely or to a large extent independently generated.**

**For precipitation, the HydroGFD3 climatology is based on the satellite-gauge merged CHPclim climatology, with some minor additions of GPCCv8 for Scandinavia and part of Siberia. Therefore CPC-Unified, GPCCv8, wfd-gpcc and wfd-cru can all be considered independent data sets and are valid for the evaluation in Fig. 5. For the daily distribution and timeseries in Figs. 7 and 9, HydroGFD3 is based on climatology with GPCCv8 anomalies, and is therefore independent of CPC-Unified, partly independent of WFDE5-CRU (besides ERA5), and to some extent independent of WFDE5-GPCC regarding the climatology.**

**For temperature, the HydroGFD3 climatology is based on CPC-Temp, and CRU is therefore independent data. The daily HydroGFD3 version for temperature is based on climatology and CRU, and is indeed strongly dependent of CPC, and partly independent of WFDE5-CRU (besides ERA5). Please note that we have corrected ourselves after your comment below on the inconsistent use of CPC and CRU for temperature.**

**The daily timeseries are plotted as a visual guide to spot issues with trends of the different data sets, with specific focus on the last years when the different tiers might affect the temporal consistency. It is not a full-fledged evaluation, but is there to give an impression of the quality of the product. Our experience from earlier studies is that in-depth evaluation can only be performed at the local scale, e.g. Fallah et al., (2020; in the reference list), and we will encourage users to pursue such evaluations.**

**About comparisons to earlier HydroGFD versions. The main purpose of HydroGFD3 is the general improvements of the methodology as well as the increased redundancy options for the realtime production. The earlier versions were discontinued with the end of ERA-Interim.**

*We have addressed this by adding a note about the independence between data to raise this issue to the attention of the reader:*

"An issue with global scale evaluations is that of independence between data sets, and most of the gauge-based data sets listed in Tab. 1 make use of more or less the same openly available observations, with regional differences. The data sets have, however, been independently generated and use different statistical models for the gridding process. Our aim is to provide a comprehensive overview of HydroGFD3 in comparison to other data sets, in order to presents its qualities and to point out potential issues. For each of the comparisons in Section 5, we chose data sets that are as independent as possible, given the limitations just discussed. Our experience from earlier studies is that in-depth evaluation can only be performed at the local scale (e.g. Fallah et al. (2020)), and we encourage users of the data set to pursue such evaluations."

2) About the long term changes in the input data sets To my understanding, any changes in the variability within or between data sets used for the bias adjustment, but also long term trends within these data sets will be transposed to HydroGFD3.0. As both the variability and long term trends might be different from one data set to another, the consecutive use of different data sets over time might induce discontinuities in the HydroGFD3.0 data. This aspect, along with a short comparison of the variability and the long term trends between consecutive data sets, should be discussed in the manuscript.

**This is indeed what we are showing in the timeseries plots. Anomalies of the data sources do contain the trends and potential discontinuities. For this version, an implicit choice was made to let trends be determined by each observational source. However, future versions could work around the issue of discontinuities by adopting the ERA5 trends and to use a shorter time window to construct anomalies of a data set.**

*We have added discussions along these lines in the revised version:*

"A future development could be to instead retain trends from the ERA5 reanalysis, and explore the use of shorter periods for calculating anomalies of the observed data. This would reduce discontinuities in the time series, but would remove the potential benefits of using trends from the observations."

Specific comments:

Section 2: It would be very useful for the understanding of the paper, if the authors could give more details about the data sets they use, e.g. better differentiate if a data set is a reanalysis or interpolated observations, the specificities of each data set and especially in which way it is complementary to or different from the other data sets. The authors should also expand the acronyms (CRU, CPC, etc.) the first time they are used.

***Thank you, we have improved the description of the different datasets along these lines.***

Section 2 : The authors talk about background climatology (l. 56), historical period (l.57, which period ?), climatological adjustment (l. 60, 61), etc. which are only explained later in sections 3 and 4. It might be easier to follow if section 2 only described the data sets used (see comment above), and if paragraph l. 56 was moved at a more appropriate place in section 3.

***Thank you, we have moved these parts to section 3 as suggested.***

l. 53 : "with a similar model as that used for ERA5". It would be useful to add some more information here (Is it the same model ? Are the biases supposed to be similar ?).

**The intention of this sentence is to justify the choice of reanalysis system ERA5 instead of other systems, such as NCEP or JRA. The similarities or differences with the operational products of ECMWF are not relevant to the topic of the paper and we do not want to confuse the reader.**

***The sentence has been re-formulated to clarify this:***

*"*This reanalysis product is chosen because our operational forecasts at SMHI (Swedish Meteorological and Hydrological Institute) are based on the medium range forecasts of ECMWF, with the same model as that used for ERA5, with similar bias, although there are differences in model version."

Section 3 : It would be interesting if the authors could add some explanation on why they calculate the correction on a monthly basis and not e.g. on a 30-day running mean basis around each day, which would smooth the correction curve applied to ERA5.

**That would have been neat, but is not possible because some of the observational data are only available at calendar month resolution.**

Section 3 : As the authors use monthly, daily, and hourly data, and also process the data on monthly, daily, and hourly time scales, it would be very useful for understanding to always clearly mention the time scale e.g. on which corrections are applied, anomalies are calculated, especially in this section, but also throughout the whole manuscript. For example, it is not clear to me whether the monthly corrections are applied to hourly or daily P.

***We have clarified the procedure of applying the scaling to each single timestep, and also that we do not currently produce hourly data. The reason is that we have not carefully evaluated the ERA5 hourly data, which is likely to have strong biases due to the coarse resolution and parameterized convection.***

l. 75: How / in comparison to which reference were the issues in CHPclim identified ?

**By visual inspection. The data set contains obvious unphysical lines for some months and regions.** *This is written out in the main text now.*

l. 80 and 82 : Is the remapping of T and Nwet also conservative as for P ?

***Thanks, we have describe this better. These two sources are mapped from 0.5 to 0.25 degree using a bilinear method for this reason.***

l. 83 : "retaining the grid points that are available consistently in all data sets and all months." Does this mean that HydroGFD3.0 data are not available for some land grid points ?

**Correct, not all land grid points are available. The grids are described in the figures and in the data set itself and most notable is that some islands are missing.**

l. 102 : The dry and wet days should be defined clearly here at latest. Is the threshold of 1 mm/day mentioned in the caption of Fig. 7 used to separate dry from wet days ?

**The method of calculating wet days does not require a definition, nor does the adjustment. Instead, the definition is imposed implicitly from the wet days in the CRU data set. This is a complicated measure since it is a mixture of instrumentation resolution and interpolation methods.**

***This has been clarified in the relevant section.***

l. 106 : Does N_clim_wet come from the CRU data set ?

***Yes, this has been written out in the revised manuscript.***

Section 3.4 : This synthesis of the correction steps is very useful. However, it might be a little bit more detailed : - To my understanding, the very first step is the preparation of the climatology. - Step 1 : "Calculate monthly anomalies in observation data" ? – Step 4 : the removal of the weakest excessive wet days should be (more clearly) explained in section 3.3. - Steps 5 and 6: This is not clear to me. Does it only concern P ? If not, I do not understand what the ratio stands for.

**Thank you for noticing the issues in the explanation. 1 is correctly interpreted. 5 and 6 is a ratio for precipitation and a difference for temperature.**

***We have revised and clarified this, also regarding the wet day correction in 3.3.***

l. 182-183 : It might be interesting to add some more discussion, e.g. remember as said before that the relative biases for P are much higher in these dry (or snowy) regions. Moreover, as no correction

is applied over Greenland and the biases are quite huge, should one conclude that the climatology is not reliable there ? It might be useful to briefly discuss the interest of not excluding Greenland as it is done for Antarctica.

**Regarding Greenland, this is a difficult topic. The observations are generally poor, and this goes for other parts of the Arctic coastline as well. Although we do not change the climatology, we still include observed anomalies for Greenland since they might provide some useful information. Antarctica is excluded because the observational data sets do not provide any information there at all.**

*We have added a few sentences to describe the issues with dry and/or snow dominated areas and the measurement uncertainty.*

"These are all dry and/or snowy regions, with an inherent observational uncertainty, adding the lower gauge network density in the areas. The presented differences between the data sets are considered well inside this expected uncertainty range. We also remark that uncertainties in Greenland are especially large due to few observations and difficult conditions, and data for this region should be used carefully, with HydroGFD3 and other data sets alike."

l. 191 : "For T, we compare to cru only, since cpct is used to build the climatology": - This is confusing as following e.g. section 4 (e.g. l. 145), fig. 3, and l. 291, cru seems to be used for the climatology, but l. 56 mentions cpct. - In any case, it might be interesting to compare with both data sets to see the impact of the bias adjustment, as it is done for P with gpcch.

*We have made the necessary clarifications in the text. cpct is used for the climatology, and cru for the anomalies in the historical period (1979—2016).*

l. 203-204 : It would be interesting to add some explanation on why the biases are larger in these regions (see also comment for l. 182-183).

*We add a sentence about the dry and cold/snowy conditions and resulting uncertainty.*

"Again, these regions have large observational uncertainty, making it difficult to determine a ground truth. "

l. 210 : Correct ". . . dry regions . . . have more dry days . . .". Are the differences really significant / worse to mention ? The values seem to be very similar.

**We mention these differences because the extreme values will be scaled to a larger extent than moderate and weak intensities, and the scaling becomes stronger when more days are removed from ERA5. A small difference in the wet days can have a large impact on the tail of the distribution.**

Paragraph l. 226 : It seems that many regions show an annual cycle. This should be briefly discussed, as it is done for T.

*We have added a short discussion of this topic:*

"All data sets show signs of an annual cycle in their anomalies to e5 in colder regions, which is indicative of differences between warm and cold season precipitation. wfde5-gpcc and wfde5-cru display stronger anomalies over the annual cycle in the colder regions compared to other data sets. This is likely due to the undercatch corrections which are larger for snowy conditions."

l. 283 : Which was the resolution of the previous versions ?

*It was 0.5 degrees, we specifically mention this in the revised manuscript.*

l. 286 : I agree and this is a major advantage of this data set. However, would such a switch between data sets not introduce biases (see e.g. the general comment on the variability and long term trends)?

**Indeed, the switch does sometimes introduce discontinuities, as shown in the timeseries plots. The use of a common climatology reduces discontinuities that arise from consistent offsets, however, it cannot affect differences in variance, or changes in the data set itself, such as changes in the underlying station network. More elaborate methods are necessary to address such issues.**

*We have added a discussion of this issue and a possible solution in the discussion section as described above.*

l. 287-288 : As section 8 follows directly, it is redundant to already mention this here.

**Indeed, it is.** *We have removed this text.*

Technical corrections:

*Thank you for the detailed reading, we have addressed the all the following points in our revision.*

l. 8 : Correct ". . . as well as the number of wet days . . ."

l. 49 : "tiers" are explained in section 4 "Data sets" not in the methodology. It might be useful to already explain here in a few words what is meant by "tiers".

l. 52 : Expand "SMHI".

Tab. 1 : Nwet acronym is not defined.

l. 71 : "climatological period" Is this 1980-2009 ?

Section 3.1 : Use data set names of Table 1 (e.g. CHPclim, GPCCv8).

l. 94 : Correct "1989-2009" to "1980-2009".

l. 128 : Correct "It happens that the land sea masks . . ."

l. 132 : Correct ". . . the output will resort no adjustment . . ." ?

l. 145 : Correct ". . . respectively for P and T."

Fig. 4 and l. 172 : With respect to which data is the bias of e5 shown here ?

Fig. 4 : It might be more consistent to represent Greenland in another way, e.g. like Antarctica, as no bias could be calculated.

l. 157 : Correct ". . . cpcp and cpct products . . ."

l. 164 : Correct ". . . cpcp and cpct products . . ."

l. 172, 206 : Correct "HydroGFD3"

l. 184 : Correct "Arabian peninsula."

l. 190 : Correct "...where both gpcch and cpcp show . . ."

l. 195 : Correct ". . . they are due to differences in elevation . . ."

l. 216 : Add reference to Figure 8.

l. 219 : Correct "Orographic effects on T were . . ."

l. 234 : Correct ". . . have similar mean . . . and show generally . . ."

l. 242 : Should it not be 2016 instead of 2017 ? If not, Fig. 3 should be adapted.

l. 244 : Correct ". . . a significant offset between the data sets . . ."

l. 245 : ". . . reduce the offset to cru." Should it not be to e5 ?

Fig. 9 and 10 : - Are these monthly anomalies ? - Correct ". . . and are evaluated . . ."

l. 258 : Correct ". . . using this method is to be able . . ." ?

l. 292 : Switch gpccm and cpct to be consistent with the previous sentence.